# Deciphering the Physicochemical and Microscopical Changes in *Ganoderma boninense*-Infected Oil Palm Woodblocks under the Influence of Phenolic Compounds

**DOI:** 10.3390/plants10091797

**Published:** 2021-08-28

**Authors:** Arthy Surendran, Yasmeen Siddiqui, Khairulmazmi Ahmad, Rozi Fernanda

**Affiliations:** 1School of Life Sciences, University of Warwick Wellesbourne, Warwick CV35 9EF, UK; 2Sustainable Agronomy and Crop Protection, Institute of Plantation Studies, Universiti Putra Malaysia, Serdang 43400, Malaysia; rozifernanda86@gmail.com

**Keywords:** basal stem rot (BSR), phenolic compounds, lignin, cellulose, hemicellulose, silica body, crystalline cellulose, biodegradation

## Abstract

The threat of *Ganoderma boninense*, the causal agent of basal stem rot disease, in the oil palm industry warrants finding an effective control for it. The weakest link in the disease management strategy is the unattended stumps/debris in the plantations. Hence, this study aimed to determine whether the selected phenolic compounds could control *G. boninense* in inoculated oil palm woodblocks and restrict wood biodegradation. Results indicated a significant reduction in the wood mass loss when treated with all the phenolic compounds. Surprisingly, syringic and vanillic acids behaved ambivalently; at a lower concentration, the wood mass loss was increased, but it decreased as the concentrations were increased. In all four phenolic compounds, the inhibition of mass loss was dependent on the concentration of the compounds. After 120 days, the mass loss was only 31%, with 63% relative degradation of lignin and cellulose, and 74% of hemicellulose and wood anatomy, including silica bodies, were intact in those woodblocks treated with 1 mM benzoic acid. This study emphasizes the physicochemical and anatomical changes occurring in the oil palm wood during *G. boninense* colonization, and suggests that treating oil palm stumps with benzoic acid could be a solution to reducing the *G. boninense* inoculum pressure during replantation in a sustainable manner.

## 1. Introduction

The rapid expansion of plantations of oil palms (*Elaeis guineensis* Jacq.) has led to its emergence as a commodity of strategic global importance. It is one of the most economically important oil crops in Southeast Asia. Palm oil is very useful, and is used extensively in food and as a precursor for biodiesel. The palm kernel cake resulting after oil extraction is also used as animal feed and bio-fertilizer. In the decades since 1980, the production of oil palm has increased from 5 Mt to more than 45 Mt, with an annual average growth rate of 7.8% [1], and is still increasing.

Trees in the tropical rain forest are hosts to a wide range of stem and root pathogens, typically belonging to the basidiomycete genera. One such predominant pathogen for the oil palm is *Ganoderma boninense*, which causes basal stem rot (BSR). Losses in the fruit bunch yield of between 0.04 and 4.34 t ha^−1^ from 10 to 22 years of planting, due to the BSR in oil palms, have been reported, and it is predicted that more than 60 million mature oil palms could be infected in Malaysia [2,3,4]. It was estimated if a tree dies at the age of 10 years, it can cause a loss of 15 years of production, which is 600 kg of crude palm oil at a monetary loss of approximately USD 675 [5].

The management of fungal stem and root rot is universally difficult [1], and BSR disease is no exception, if not more difficult than most. This fungus is a saprophyte that can survive for a long time in the debris, stumps and leftover roots of unattended logs in the plantation [1]. The initial spread of BSR in oil palm could be via basidiospore dispersal, root to root contact, and also via root contact with the unattended infected debris [6,7]. The unattended debris acts as a source of the inoculum of *G. boninense* during the replantation. The reports suggest that the disease incidence (DI) has been increasing in successive replantation. It has been reported that in the first replantation, oil palm plants of 10 years old have been affected with less than 2% of DI; however, at the third replantation, DI (70%) has increased alarmingly [8]. In his study, Khairudin (1993) has concluded that 93% of seedlings that are growing around the diseased stumps have also been infected with BSR disease [9].

The newly cut stumps and the unattended infected trunks are the most vulnerable links in the disease management chain. This is because the fresh-cut stumps and the trunks are still alive, and serve as a nutrient reservoir for the pathogens to flourish. The treatment of the cut stumps with chemicals is not effective because of the larger size and the anatomy of oil palm roots. Oil palm roots form around an orthotropic taproot, and the horizontal lateral roots occupy about 16 m^3^ of soil mass [10], which makes it very tedious and expensive to excavate as a way to control BSR disease.

To control BSR disease in oil palms, the *G. boninense* surviving in cut stumps must be eradicated. To protect the surface of the stump from attack by *G. boninense*, there is an urgent need to find some antimicrobial compounds that can travel to the site of infection (roots) efficiently and act cost-effectively. Through a series of experiments and screening, four naturally occurring phenolic compounds, namely, benzoic, salicylic, syringic and vanillic acids, were selected based on their antifungal efficiency, along with their inhibitory potential toward the ligninolytic and cellulolytic enzymes [11,12]. These naturally occurring phenolic compounds are involved in the lignin synthesis pathway in oil palm and hence could be potential agents to eliminate *G. boninense*. To the best of our knowledge, little or no study has been conducted to evaluate the degradation pattern of *G. boninense* in the presence or absence of selected phenolic compounds, which could be important to our understanding for the development of an alternative BSR management technique.

## 2. Materials and Methods

### 2.1. Chemicals

Potato dextrose agar (PDA), salicylic acid, syringic acid and vanillic acid were purchased from Friedemann Schmidt (Germany), while benzoic acid was obtained from R&M Chemicals and Reagents (Malaysia). Four phenolic compounds with a concentration of 1, 5, 10 and 15 mM were used.

### 2.2. Microorganism, Culture Conditions and Treatments

The isolate *Ganoderma boninense* (PER 71) was obtained from the Malaysian Palm Oil Board (MPOB). The culture was maintained by sub-culturing the fungus at regular intervals on PDA at 28 °C. Four phenolic compounds, namely, benzoic acid, salicylic acid, syringic acid and vanillic acid, were tested to see whether they would inhibit the biodegradation ability of *G. boninense* on oil palm woodblocks.

### 2.3. Biodegradation of Woodblocks

The rate of woodblock decay was investigated using the method described by Schirp and Wolcott (2005), with slight modifications [13]. The healthy oil palm trunks were collected from the MPOB, and were cut into 20 × 20 × 40 mm blocks. The blocks were oven-dried at 60 °C for 48 h to obtain a constant dry weight. The dried woodblocks were weighed and labeled individually, to obtain the initial dry mass, followed by double sterilization for 30 min at 121 °C, as described by Bucher et al. (2004) [14]. Tissue-culture jars (350 mL capacity, 15 cm in length and 75 mm in diameter) containing 20 mL of PDA media were autoclaved at 121 °C for 20 min, followed by cooling and solidification of the media. The jars were then inoculated with *G. boninense* culture (two mycelial plugs of 3 mm Ø) and incubated at 28 ± 2 °C until full coverage of the media by fungal mycelium was reached. The phenolic stock solutions were prepared according to Sudrendran et al. (2017) [11]. The sterilized woodblocks were dipped for 30 min in the respective phenolic compounds of each concentration (1, 5, 10 and 15 mM) and air-dried in the laminar hood until no dripping was observed [14]. After complete colonization of PDA by *G. boninense* in the jars, the sterilized healthy and phenolic compound-treated oil palm woodblocks were placed individually on the top of the *G. boninense* culture. The jars containing woodblocks without any phenolic compound treatment served as the controls. All the jars were then incubated at 28 ± 2 °C for 10, 30, 45 and 120 days. For each sampling period, five woodblocks were harvested. The surface of each block was carefully wiped to remove any mycelium, followed by weighing. The experiments were repeated twice, and the mass loss was calculated using Equation (1):(1)$$Percentage\;mass\;loss\;\left(\%\right)=\frac{Initial\;weight\;after\;treatment-Final\;weight}{Initial\;drymass}\times 100$$

### 2.4. Anatomical Characterization during Biodegradation of Oil Palm Woodblocks

#### Scanning Electron Microscopy (SEM) Analysis

At the end of each incubation time, the *G. boninense*-colonized wood samples were collected. The mycelia from the samples were wiped off. The radial and vertical samples, with a dimension of 0.5 × 0.5 cm, were sputter-coated with gold–palladium alloy. The SEM (S-3400N Hitachi, UPM, Serdang Malaysia) was performed to determine the morphological changes of the wood and *G. boninense* during the degradation period.

### 2.5. Chemical Characterization during Biodegradation of Oil Palm Woodblocks

#### 2.5.1. Fourier Transform Infrared (FT-IR) Spectroscopy Analysis

At the end of each incubation period, the *G. boninense*-colonized woodblocks were sampled out of the jars, and the mycelium was carefully wiped off from the woodblocks. Later, the woodblocks were impregnated with water to remove the remaining mycelium. The wood samples were powdered in the mortar and pestle using liquid nitrogen. The powder was sieved, and those fractions with an average size less than 0.5 mm were retained to develop KBr pellets. The FTIR spectra for the wood samples were measured by direct transmittance using the KBr pellet technique. The amount of sample in the pellets was constant (5 mg/500 mg KBr). The spectra were recorded in the range of 4000–400 cm^−1^ using a Spectrum 100 FTIR Spectrometer (Perkin Elmer, Inc., Waltham, MA, USA). Five recordings were performed for each sample, and evaluation was made from the average of the five using the software [15]. Healthy wood without any treatment was also analyzed to compare with the degraded wood. From the FT-IR spectra, the lower order index (LOI, A_1430_/A_898_) was used to estimate the crystal structure of cellulose material [16], and the total crystallinity index (TCI, A_1372_/A_2900_) was used to estimate the infrared crystallinity index [16]. The ratio between the syringyl and guaiacyl S/G (S/G, A_1327_/A_1271_) units in the lignin was also estimated [17].

#### 2.5.2. Thermogravimetry (TGA) Analysis

At each sampling time, the wood samples were removed from the jars and cleaned of mycelium, as mentioned above. The wood samples were then powdered, using a mortar and pestle and liquid nitrogen. The powder was sieved; the fraction with an average size of less than 0.5 mm was retained. TGA analysis was performed using a Perkin Elmer thermogravimetric analyzer (Pyris 1 TGA, Pyris software 7.0—Perkin Elmer) at a constant nitrogen flow of 20 mL min^−1^ and constant heating of 15 °C min^−1^. The heating scans were performed with 5 mg of wood samples in the temperature range of 25–890 °C [15]. Healthy wood without any treatment was analyzed for comparison with the degraded wood. The amount of lignin, cellulose and hemicellulose was quantified according to Yang et al. [18]. Oil palm lignin degradation takes place at 700–800 °C, hemicellulose and cellulose at 0–300 °C and 330–340 °C, respectively. The relative percentage of lignin, cellulose and hemicellulose was calculated according to Equation (2):(2)$$Relative\;lignin\;degraded\;\left(\%\right)=\frac{Initial\;lignin\;content-Final\;lignin\;content}{Initial\;lignin\;content}\times 100$$

For the relative percentage of cellulose and hemicellulose when degraded, the lignin in Equation (2) is replaced with cellulose and hemicellulose, respectively.

### 2.6. Statistical Analyses

The data were analyzed using SAS statistical software (PC-SAS software V8.2, SAS Institute, Cary, NC, USA). A *p*-value of ≤0.01 was considered significant. All the experiments were repeated twice, with six replicates for each treatment. The means were compared using Tukey’s range test.

## 3. Results

### 3.1. Mass Loss

Initially, the colonization of *G. boninense* in the control woodblocks was very slow and steady for the first ten days. Eventually, *G. boninense* rapidly colonized the control woodblocks after 30 days, with a weight loss of 37%. With the progression of the incubation period, a significantly higher (*p* ≤ 0.05) weight loss was observed in the control woodblocks, compared to the woodblocks treated with 1 mM benzoic acid. The mass loss at various time intervals, i.e., 10, 30, 45 and 120 days, are shown in Figure 1a–d, respectively. As the colonization of *G. boninense* increased with time, a decrease in the weight of the oil palm woodblocks was observed. The colonization rates in those treated woodblocks with a higher concentration of phenolic compounds were prominently less when compared to the control woodblocks, except for the syringic and vanillic acid treatments. On the 120th day, the mass loss due to the colonization of *G. boninense* on the woodblocks that were treated with 1 mM of syringic (74.9%) and vanillic (83%) acids was more marked when compared to the control (71.8%) woodblocks. These two phenolic compounds enhanced the growth of *G. boninense* at their lower concentration, that is, 1 mM. However, the same compounds proved to be poisonous when the concentrations increased further. As the concentration of the syringic acid was increased to 5 mM in the woodblocks, a significant decrease in mass loss due to the colonization of *G. boninense* was observed. The weight losses in the woodblocks treated with 5 mM syringic acid (25.5%) were almost half when compared to the 1 mM syringic acid treatment (Figure 1d), whereas, on the 120th day, the 10 mM vanillic acid-treated woodblocks had a significant reduction in weight loss of about 19.6%. At the end of the study, among the four phenolic compounds, benzoic acid was the best inhibitor against *G. boninense* colonization at 1 mM concentration, followed by salicylic acid, with a 40.3% weight loss in the treated woodblocks. The woodblocks treated with 5 mM and above of benzoic acid, 10 mM and above of salicylic and syringic acids, and 15 mM of vanillic acids retained their physical properties, with no growth of *G. boninense,* as in the healthy oil palm woodblock, until the end of the study. At the end of the decay period, the degraded wood turned from brown-black to whitish tan with some black spots. The woodblocks were soft, fragile and easily shredded into longitudinal fragments.

### 3.2. Anatomical Characterization during Biodegradation of Oil Palm Wood

#### Scanning Electron Microscopy (SEM) Analysis

The SEM micrographs of the healthy oil palm woodblocks revealed many silica bodies embedded on the surface of the fiber strands. The silica bodies were attached to circular craters and were arranged in a pattern of flowers that typifies oil palm wood (Figure 2a). Silica bodies were spread uniformly over the strand’s surface. In general, the SEM observations indicated that, initially, *G. boninense* colonized the substrate surfaces via apical hyphal extension (Figure 2b). The initial colonization on the 10th day was up to the cell lumen, which provoked erosion. As the degradation progressed, the eroded zones coalesced, and the fungal hyphae filled in the voids at 30 days. At the end of the degradation (120 days), silica bodies on the surface of the oil palm woodblocks were completely dissociated, and extensive deterioration of the wood cell walls was observed in the control (Figure 2c).

The SEM of the woodblocks treated with 1 mM vanillic and syringic acid displayed a similar pattern, with denser hyphae than the control (Figure 2d). Although flower-shaped silica bodies were surrounded by the *G. boninense*, the silica bodies looked intact (Figure 2d). After incubation for 30 days, the hyphae of the *G. boninense* were abundant and were spread across the pits of the vessels, as well as the fibers, in a bridge-like arrangement. As the degradation time was prolonged, very few boreholes were observed in some of the parenchymal tissue, and *G. boninense* intervention was evident (Figure 2e). At the end (120 days) of this study, the woodblocks treated with 1 mM vanillic and syringic acids displayed more disintegrated parenchymal tissues, compared to the control (Figure 2f). No flower-patterned silica bodies were observed in the woodblocks treated with 1 mM of syringic and vanillic acids. Similar observations were also observed in the woodblocks treated with 5 and 10 mM vanillic acid, and also in those treated with 5 mM syringic acid, but the intensity of the colonization was less marked. No *G. boninense* growth was observed in the woodblocks treated with 10 and 15 mM syringic acid and the 15 mM vanillic acid-treated woodblocks. The woodblocks treated with 1 and 5 mM salicylic acid were colonized with *G. boninense,* but with low visible intensity. However, the woodblocks with 5 mM concentration were significantly better, with less colonization. The colony of hyphae in 5 mM salicylic acid-treated woodblocks displayed a cleft circle at the end of the hyphae and, after a closer look, it was confirmed that the hyphae were almost emptied (Figure 2g,h). No growth of *G. boninense* was observed on the woodblocks treated with a 10 mM concentration of salicylic acid. *G. boninense* colonized the least heavily in the woodblocks treated with 1 mM benzoic acid until the end of the study. *G. boninense* hyphae were thin on the 10th day, when compared to the *G. boninense* grown on the control woodblocks. At the end of the degradation process (120th day), dense colonization was observed, with very thin hyphae of *G. boninense* on those woodblocks treated with 1 mM benzoic acid (Figure 2i); the flower-shaped silica components embedded in the circular craters were intact on the surface of the woodblocks.

### 3.3. Chemical Characterization during Biodegradation of Oil Palm Woodblocks

#### 3.3.1. Fourier Transform Infrared (FT-IR) Spectroscopy Analysis

FT-IR spectroscopy was performed to compare the changes in the functional groups in the wood components in both healthy and treated wood samples that have been degraded by *G. boninense*. This study’s focus was on the band modifications related to the carbohydrates (cellulose, hemicellulose and starch) and the lignin in the oil palm wood. The FT-IR spectra in the fingerprint region between 1800 to 500 cm^−1^ were analyzed. Appendix A represent the LOI, TCI and S/G of the wood components. The TCI represents the crystallinity degree of the cellulose, and LOI represents the overall degree of order in the cellulose. This analysis has given a clear demonstration of the structural degradation of cellulose. Not many changes were observed in the first ten days of the degradation process between the control and the phenolic compound-treated woodblocks. As the samples were degraded by *G. boninense,* a 3340 cm^−1^ band increased in its intensity and broadening. This was observed in both the control and the treated samples. As the band broadened, it shifted to the lower frequency. However, the intensity of this band in the wood samples that were treated with 10 mM vanillic, 5 mM salicylic, and 1 mM of benzoic acids was less intense when compared to the control samples. The TCI of the control samples increased until the 30-day point and started decreasing as the degradation period continued, but the LOI remained constant for the control samples for the first 30 days and decreased as the degradation continued (Appendix A). However, woodblocks treated with 1 mM benzoic acid maintained almost the same LOI until the end of the study, but a constant decrement was observed in the TCI values. The LOI values of the woodblocks treated with salicylic acid were constant for the first 30 days, and decreased with the increase in the degradation period. On the other hand, the TCI values of the woodblocks treated with 1 mM salicylic acid decreased for the first 30 days, and started increasing until the 45th day, again decreasing from then onward. The LOI and TCI values of the wood samples treated with vanillic acid continued reducing throughout the degradation period, except for the TCI values on the 45th day.

As shown in Table 1, *G. boninense* invaded the lignin, cellulose and xylene bonds simultaneously. Only stretching in various bonds was observed, and no bending was visible in the FT-IR analysis. The stretching of the C-O bonds of the syringyl and guaiacyl units of lignin, and also the stretching of the C=O bonds of aryl ketone and C=C in the aromatic group of lignin, occurred on the 30th day. Besides lignin, C-H and C-O stretching in cellulose and xylene were detected. A band at 1700 cm^−1^ was observed in the wood samples treated with 1 mM benzoic, 5 mM salicylic and 15 mM vanillic acids. This band is associated with the C=O stretching of conjugated or aromatic ketones (1700 cm^−1^) [15] at days 30 and 45. As degradation progressed further, this peak disappeared (day 120). The intensity of the band at 1670 cm^−1^ corresponds to the C=O stretching of lignin, which is more intense in the samples treated with 1 mM syringic and vanillic acids. The LOI and TCI values also dropped. This indicates the extensive degradation of cellulose and hemicellulose components, which increased the residual lignin concentration. These findings were in agreement with the TGA studies (results below). Similar bands appeared for the remaining treatments, with lesser intensity. In the case of 1 mM benzoic acid, as the degradation progressed further, the band at 1670 cm^−1^ disappeared. A constant decrement was observed in the S/G ratio in the control wood samples. This indicates that *G. boninense* degraded more G units in the lignin compared to the S unit. In only the wood samples treated with syringic and vanillic acids, the S/G ratio has slightly increased by the 30th and 45th days, and dropped at the 120th day of biodegradation.

#### 3.3.2. Thermogravimetry (TGA) Analysis

The chemical composition of healthy oil palm woodblocks that were treated with phenolic compounds and infected with *G. boninense* are shown in Table 2 and Appendix A. The three distinct stages of weight loss were observed in both the treated and the control oil palm woodblocks. In TGA, the first is the dehydration stage, followed by active pyrolysis and passive pyrolysis; these were in the ranges of 0–100 °C, 100–300 °C and 350–800 °C. A negligible mass loss was observed both in the control and in the treated samples at the dehydration stage. A considerable mass loss had occurred in between the temperature ranges of 110–500 °C. The degradation of the cellulose and hemicellulose had occurred in this range, which was the start of the active pyrolysis step. The estimation of hemicellulose, cellulose and lignin were recorded at temperature ranges of 220–300 °C, 330–340 °C and 700–800 °C, respectively [18]. As the degradation progressed further, an increase in the thermal degradation rate was observed in the woodblocks colonized with *G. boninense*. Appendix A represents the lignin, cellulose and hemicellulose contents of the healthy oil palm woodblocks before the treatment.

The relative percentages of lignin, hemicellulose and cellulose degradation in control woodblocks colonized with *G. boninense* were elucidated according to TGA; on the 10th day, *G. boninense* had degraded the hemicellulose (68.8%), followed by lignin (49.6%), and then cellulose (37.1%). As the degradation progressed further, the rate of degradation of the hemicellulose had decreased gradually and, simultaneously, the rate of degradation of the lignin had increased in parallel on the 120th day. Hence, at the end of the degradation period (120th day), *G. boninense* had degraded the lignin and hemicellulose almost equally, but at the same time, the degradation in the cellulose was the least. This result indicated that *G. boninense* had utilized all the three components of the oil palm wood during colonization. During the initial stages of colonization, *G. boninense* preferred more hemicellulose components to degrade, but at a later stage, there was a change in preference toward lignin rather than hemicellulose.

Table 2 indicates the difference between the treated woodblocks with that of the control woodblocks, and the relative degradation of lignin, hemicellulose and cellulose components. An increase in the concentration of phenolic compounds in the treatment woodblocks had decreased the degradation rate, as informed by the lesser colonization of *G. boninense* and the lesser utilization of carbon sources.

The maximum destruction of lignin was recorded in the woodblocks that were treated with 1 mM vanillic acid, and this tendency continued until the end of the study. On the 10th day of colonization of the *G. boninense* on oil palm woodblock treated with 1 mM vanillic acid, degradation ratios of 62.4, 64.9 and 40.5% of lignin, hemicellulose and cellulose, respectively, were recorded. This was followed by 1 mM syringic acid-treated woodblocks, and on the 10th day, 54.9% of the lignin, 49.2% of the hemicellulose, and 34.8% of the cellulose components were found to have degraded. On the 45th day, 72.2% of the lignin was found to have degraded in the 1 mM vanillic acid-treated woodblock, as against 64.7% of lignin degradation in the control woodblocks. The degradation pattern in oil palm wood treated with vanillic acid appeared similar with hemicellulose and lignin to that of the control woodblocks, initially, but the destruction of lignin was found to be discernibly higher in the 1 mM vanillic acid-treated woodblocks (Table 2). However, in the case of woodblocks treated with syringic acid, *G. boninense* degraded more lignin and cellulose components than it did the hemicellulose components.

Overall, the woodblocks treated with 1 mM benzoic acid recorded a minimum degradation rate when compared to other phenolic compounds at the minimal concentration. The degradation in those woodblocks treated with 1 mM benzoic acid was accelerated after the 10th day. The lignin degradation had increased from 2.3% on day 10 to 54.95 on day 30; similarly, hemicellulose destruction increased from 18.7% to 55.5%, and in the case of cellulose, from 14.4% to 44.5%, respectively. At the end of this study, the relative percentages of lignin, hemicellulose and cellulose that had degraded in the woodblocks treated with 1 mM benzoic acid were 63.2%, 63.9% and 74.2%, respectively. Slow and steady rates in the degradation of wood components were observed in those woodblocks treated with 1 and 5 mM salicylic acid, but the ratio of intensity varied between them temporally. On day 10, the ratios of the destruction of lignin, hemicellulose and cellulose were at 47.4–32.3%, 33.4–22.4% and 15–9%, respectively. On the final day, *G. boninense* degraded a higher percentage of lignin, cellulose and hemicellulose, at 72.2%, 60.9% and 71.1%, respectively.

## 4. Discussion

Abiding by the “zero burnings” policy, instead of controlled burning, the felled oil palm trunks and the stumps (with roots) are left to degrade in field conditions. This serves as a host for the saprophytic, pathogenic *G. boninense* to survive for a long time until they come in contact with healthy oil palm roots. Increases in disease incidence and disease severity were observed across the consecutive replantation. This is directly linked to the increase in size of the inoculum of *G. boninense* [30]. Nearly 60% of the estates in Southeast Asia were reported for the presence of this disease [1]. Although the stumps and felled oil palm trunks are considered to be the weakest link in the BSR disease management strategy, it is one that has been overlooked for some time.

Recently, researchers have initiated research on the identification of biocontrols of *G. boninense* in the felled trunks [31]. Despite the theoretical fact that biocontrol agents are superior to other control agents, their efficiency in the field is always questionable. This issue apart, the introduction of new biocontrols into the environment can also cause an unwanted change to the biodiversity of the ecosystem [25]. Hence, to overcome the above-mentioned issues, naturally occurring phenolic compounds might be a potential candidate to control the *G. boninense* that resides saprophytically in the oil palm trunks.

Hence, an in vitro study was conducted to evaluate the effects of phenolic compounds on controlling the growth of *G. boninense* in oil palm woodblocks. The wood can be studied in the following three ways, microscopic, spectroscopic, and thermodynamics, providing useful information on the surface morphology and chemistry of wood samples during their degradation by *G. boninense* when in the presence of phenolic compounds. In the current study, the analyses were made using SEM, TGA and FTIR.

*G. boninense* is a white-rot fungus (WRF) that is known for its remarkable ability to produce both oxidative and hydrolytic enzymes for the degradation of lignin and other cellular components [32]. In our study, the mass loss at the end of the degradation by *G. boninense* was 70% on the 120th day. The observed mass loss in this study is 10–20% higher than the *Populus deltoides* wood degraded by *Pycnoporus sanguineus* and *Ganoderma. lucidium* at 120 days [24]. Both *P. sanguineus* and *G. lucidium* are considered to be extensive degraders of poplar wood. From the point of view of mass loss, we can state that *G. boninense* can be potentially destructive under favorable conditions. Although this in vitro study cannot be considered as absolute evidence of the behavior of *G. boninense,* still, it is useful to predict the trajectory of degradation of oil palm wood.

During the degradation period, black spots appeared on the woodblocks, which could be associated with the deposition of manganese peroxidase by the fungus that colonizes the woodblocks [20]. *G. boninense* is known to produce three ligninolytic enzymes, namely, laccase, lignin peroxidase and manganese peroxidase [12]. At the end of the degradation process, the woodblocks were soft, spongy and pale in color, which is the characteristic feature of wood degraded by WRF [33].

In this study, a higher concentration of phenolic compounds was observed to reduce the growth of *G. boninense* on oil palm woodblocks. The rate of mass loss due to the colonization of *G. boninense* is inversely pronominally connected to the concentration of the phenolic compounds. However, the two para-methylated phenolic compounds (syringic and vanillic acid) behaved ambivalently, and at the lower concentration of phenolic compounds, increased the colonization of *G. boninense,* thus leading to increased mass loss. A higher mass loss of about 80% and 71% was also observed in the woodblocks treated with phenolic compounds, at 1 mM of the vanillic and syringic acids, respectively. This behavior is not surprising because these two phenolic compounds are not only metabolized by *G. boninense* but also by a wide range of white-rot fungi (WRF), such as *Phanerochaete chrysoporium, Prognathodes dichrous* and *Pleurotus ostreatus,* at lower concentrations [28,34]. These para-methylated aromatic compounds are less inhibitory to the growth of the WRF, as well as being easy to degrade. The conversion of phenol into para-methylated forms is considered to be one of the stratagems of the WRF while colonizing lignocellulose materials [35].

*G. boninense* produces the laccase enzyme [12], which acts as a detoxifying enzyme, as well as being responsible for the degradation of aromatic structures in the lignin. Hence, the phenolic compounds at lower concentrations in this study would have been oxidized by the laccase enzymes produced by *G. boninense*. The enhanced laccase enzyme produced by *Coriolus versicolor* has also detoxified various phenolic compounds, such as ferulic acid, xylidine, vanillic acid, cinnamic acid and guaiacol at 1 mM, but, as the concentration increased further, the phenolic compounds inhibited the growth of *C. versicolor* [36].

The presence of phenolic compounds at minimal concentration, along with the oil palm wood chips in the broth, not only increased the production of oxidative enzymes but also hydrolytic enzymes [11,12,37]. From this, we can infer that the phenolic compounds at lower concentrations induced the production of the oxidative and hydrolytic enzymes; as a result, more substrate (lignin, cellulose and hemicellulose) has been utilized by the enzymes [11,37]. This, in turn, led to a substantial mass loss in the oil palm woodblocks.

When the wood samples were viewed under SEM, the images indicated that *G. boninense* is a WRF that adapts to a simultaneous degradation pattern; that is, the degradation of the lignin and cellulose components of the oil palm woodblocks. The wood cell walls are attacked from the lumen, and the degradation is associated with the hyphae and not with the diffusion mechanism [33]. The erosion troughs observed beneath the hyphae indicate that *G. boninense* adopts a simultaneous degradation of lignin and cellulose components. This result was similar to *G. lucidium,* which degrades both lignin and cellulose [24]. Some of the species of WRF are known for their ability to degrade the wood components simultaneously. The rate of lignin and cellulose compound removal depends on various factors, such as the type of wood substrate, as well as the fungal species [21]. The surface recalcitrance of healthy oil palm wood was evident in the micrographs. The silica bodies serve as a major physical barrier for the enzymatic hydrolyzing process, due to the difficulty in penetrating the wood surface to access the cellulose and hemicellulose for sugar production. Although benzoic acid-treated wood at lower concentrations could not inhibit the *G. boninense* growth, it prevented the disintegration of silica bodies and could have aided in strengthening the structural barrier. The presence of phenolic compounds did not relatively alter the degradation pattern; however, it modified the extent of colonization by *G. boninense* on the surface of the woodblocks.

In the process of degradation, the WRF considerably affects the lignin structure. The demethoxylation of lignin takes place at the initial processes of degradation, followed by the formation of oxidized products [22]. This makes the lignin even more complex to detect [38]. Hence, FT-IR and TGA analyses were performed to further analyze the chemical changes in the wood structure during the biodegradation, as well as to quantify the remaining wood components. In FT-IR spectroscopy, the modification occurs in the cellulose as well as the lignin components of the oil palm woodblocks during the degradation process. A great reduction was observed at 1410 and 1415 cm^−1^ (cellulose), 1375 cm^−1^ (cellulose and hemicellulose) and also at 1028 cm^−1^ (starch) [29]. As the degradation proceeded further, these peaks disappeared. The peaks corresponding to the lignin (166, 1634, 1628 cm^−1^) had increased initially and later disappeared, as was evident in the current findings [19,20]. Again, these suggest that *G. boninense* is a simultaneous degrader of lignin, hemicellulose and cellulose components, almost equally. When comparing the wood components, the highly crystalline cellulose components are the least degraded. This could be due to the presence of the strong covalent bonds between them, as well their interaction with the lignin [23].

The FT-IR spectra showed that the crystallinity indexes (TCI, LOI) increased initially and then decreased. The initial increment and later decrement in the crystallinity index were observed in various wood-rotting fungi [39]. This indicated the higher accessibility of the cellulase enzyme initially; thus, the substrate (cellulose) has degraded, leading to a decrease in the crystallinity index [40]. As the degradation of oil palm wood continued, the ordered crystalline structure of the cellulose was disturbed, and more amorphous domains were also introduced. The S/G ratio constantly decreased throughout the study of the control, indicating that *G. boninense* preferred S-type lignin more than the G-type. This could be due to the high prevalence of ether-type linkages, compared to G-type lignin [26]. However, the S/G ratio increased in all the woodblocks treated with syringic and vanillic acid from 30 to 45 days, and a great decrement was observed on the 75th day. This indicates that the presence of phenolic compounds in the oil palm wood can influence the preference of the type of lignin that *G. boninense* utilizes.

According to our previous study, the predominant oxidative enzyme of *G. boninense* is laccase [12]. Peroxidases (lignin and manganese) are produced predominantly by *G. boninense,* but no significant difference was observed in the production of hydrolytic enzymes in the presence of syringic and vanillic acids [12,37]. The results revealed that the presence of phenolic compounds only alters the preference of lignin subunits (S and G) and not the degradation pattern or the degradation ratio of lignin-hemicelluloses and cellulose. However, an increase in the concentration of phenolic compounds reduced the quantity of the degraded components.

The phenolic compounds are the key compounds in signal transduction in the plant defense system, and are also involved in the elimination of pathogens. Hence, an increase in the content of phenolic compounds in the plant is linked to induced resistance in plants [27,41]. The phenolic compounds used in this study are associated with lignin and lignin biosynthesis pathways. Syringic and vanillic acids are involved in the lignin structure, whereas benzoic acid and salicylic acids are involved in the synthesis of lignin. The cells of cut stumps and the felled trunks are intact and alive for a considerable period. Hence, the phenolic treatment of felled oil palm trunks and stumps can control *G. boninense* and can develop resistance against it.

A three-year study conducted by the Malaysian Palm Oil Board (MPOB) on live BSR-infected oil palm stumps treated with 1.2 kg Dazoment revealed a 10% survival rate of *Ganoderma* inoculum [42]. This practice can greatly reduce the inoculum survival rate. Nevertheless, the cost of the amount of Dazoment required to eradicate the inoculum should not be overlooked. The cost of the above-mentioned phenolic will comparatively be less, and be used in reduced quantities to treat the palm; hence, it could be a cost-effective method. Besides this, the salient feature of phenolic compounds is that they are resistant to autoxidation [43]. Hence, they could be directly applied to the stump via soil drenching. However, the effect of these phenolic compounds on the environment should be evaluated before their application in the field.

## 5. Conclusions

The results of this study verify and conclude that naturally occurring phenolic compounds can be an effective controller of BSR disease, and, hence, they are considered to be potential compounds to treat the infected stumps and debris. Benzoic acid is the most effective compound to control *G. boninense* growth in the wood samples at the minimal concentration (5 mM), followed by salicylic acid. As a recommendation, it is essential to determine the threshold concentration of phenolic compounds before their application, because a lower concentration of phenolic compounds has enhanced the colonization of *G. boninense,* which defeats the purpose of treatment. This study could serve as a stepping-stone towards the development of an effective stump treatment, with the ultimate aim of minimizing BSR infection in oil palm estates in a sustainable manner.

## Figures and Tables

**Figure 1 plants-10-01797-f001:**
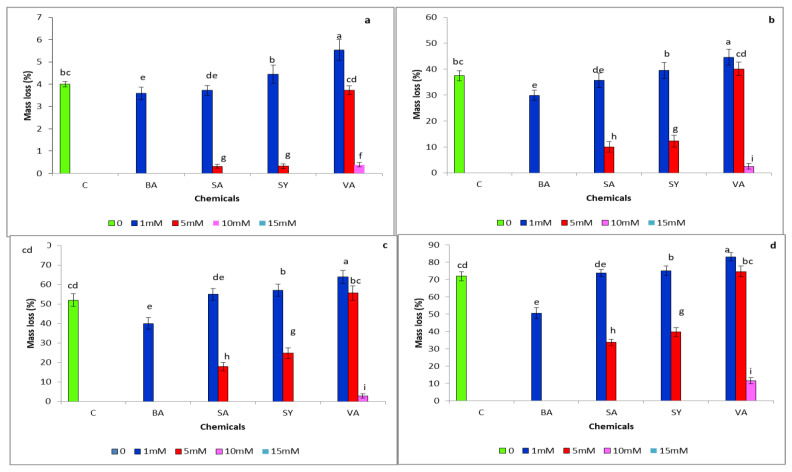
Degradation of oil palm wood by *G. boninense* with and without phenolic treatments at various time intervals. (**a**) 10th-day sampling of degraded oil palm wood; (**b**) 30th-day sampling of degraded oil palm wood; (**c**) 45th-day sampling of degraded oil palm wood; (**d**) 120th-day sampling of degraded oil palm wood. Vertical bars indicate standard deviation (S.D.) *n* = 6. C—control, BA—benzoic acid, SA—salicylic acid, SY—syringic acid and VA—vanillic acid. No data due to absolute inhibition for BA (5, 10 and 15 mM), SA (10 and 15 mM), SY (10 and 15 mM) and VA (15 mM).

**Figure 2 plants-10-01797-f002:**
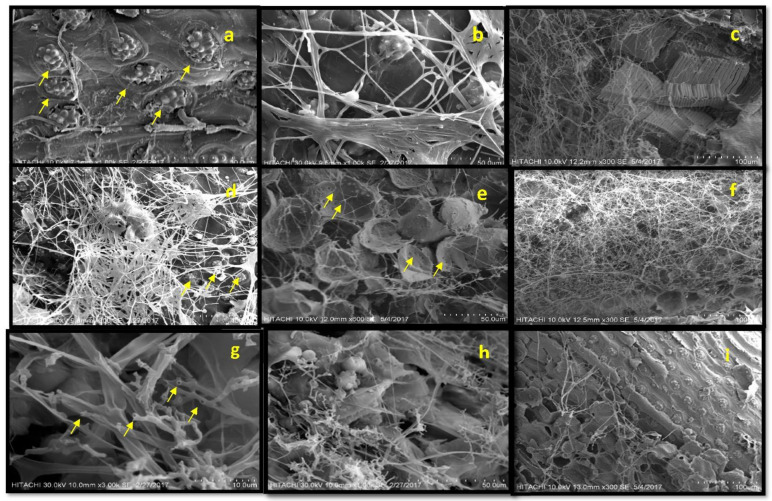
Scanning electron microscope (SEM) analysis of healthy and infected oil palm wood sample subjected to various treatments. (**a**) Intact silica bodies in healthy oil palm wood; (**b**) the initial stage of apical colonization of *G. boninense* in oil palm wood; (**c**) the control wood samples with silica bodies on the surface of oil palm woodblocks were completely dissociated, and show extensive deterioration in the wood cell walls; (**d**) dense hyphae on oil palm wood treated with 1 mM vanillic acid; (**e**) woodblocks treated with 1 mM syringic acid at 30th day; (**f**) colonization of *G. boninense* on woodblocks treated with 1 mM vanillic acid at the120th day; (**g**) a closer look at the hyphae with clefts grown on woodblocks treated with 5 mM salicylic acid; (**h**) *G. boninense* with deformities on woodblocks treated with 1 mM salicylic acid; (**i**) intact silica bodies on the oil palm woodblocks treated with 1 mM benzoic acid at the 120th day.

**Table 1 plants-10-01797-t001:** Characteristic bands in the FT-IR spectra of the studied oil palm wood samples in the 400–4000 cm^−1^ region.

Wave Number (cm^−1^)	Assignment	Source	Observation	References
Lignin Bands				
1700	C=O stretching of conjugated or aromatic ketones	Lignin	Appeared in wood samples treated with 1 mM benzoic, 5 mM salicylic and 15 mM vanillic acids. Later disappeared on the 120th day.	[19]
1612	Unconjugated carboxyl stretch of both lignin and cellulose	Lignin	A high-intensity band appeared in woodblocks treated with 1 mM syringic and vanillic acids. A medium-intensity band was observed in 1 mM benzoic acid initially, and was not observed in the later stages of degradation.	[20,21]
1620	C=C stretching in the aromatic groups of lignin	Lignin	This band was observed in all the treated as well as the control wood samples.	[21,22]
1670	C=O stretching in the conjugated p-substituted aryl ketone	Lignin	This band was observed in all the treated as well as the control wood samples.	[19,23]
1270	C-O stretching in xylene and hemicellulose and guaiacyl structure in lignin	Lignin	The intensity of the band decreased when the lignin and the adjacent hemicellulose degraded appeared in the control wood samples, along with wood samples treated with 1 mM phenolic compounds. It appeared in all the other wood samples as the degradation proceeded.	[24]
1034	Deformation vibration of C-H bond in aromatic rings	Lignin	This band was observed in all the treated as well as the control wood samples.	[25]
1247	C-O stretching in lignin (Guaiacyl units) and hemicellulose	Lignin	This band was observed in all the treated as well as the control wood samples. However, the highest intensity of this band was observed in wood samples treated with 1 mM vanillic acid.	[19,23]
**Carbohydrate Bands**				
1336	OH in-plane bending cellulose	Cellulose	A weak band appeared only at the end of the degradation period.	[26,27]
1320	C-H variation in cellulose and C-O stretching in syringyl unit of lignin	Cellulose and lignin	The intensity of the band decreased when the lignin and the adjacent hemicellulose degraded.	[28]
128	C-H stretching	Starch	A weak band appeared only at the end of the degradation period.	[14]
**Hydroxy Bands**				
3427	Intra-molecular OH stretching in cellulose	Cellulose	Increased intensity of this band indicates that more hydroxyl groups are available resulting from the hydrolysis. The increment in these two bands was observed throughout the degradation process.	[29]
3340	Bonded OH stretching	Carbohydrate and lignin		

**Table 2 plants-10-01797-t002:** Percentage of the chemical composition of oil palm wood (with and without treatment) degraded by *G. boninense* at various time intervals.

	Biodegradation (Days)											
Phenolic Compounds (mM)	10th Day			30th Day			45th Day			120th Day		
	Relative (%) Degradation											
	L	C	H	L	C	H	L	C	H	L	C	H
Control	49.6	37.1	68.9	55.6	46.5	69.9	64.7	46.5	70.6	72.2	60.9	71.1
BA 1	2.3	14.4	18.7	54.9	44.5	35.5	56.4	47.5	49.2	63.2	63.9	74.2
SA 1 SA 5	47.4	15.1	33.4	54.9	40.1	48.5	57.1	49.5	66.6	72.2	61.9	70.6
	32.3	9.0	22.4	45.7	27.1	39.8	49.3	29.8	41.1	68.0	35.8	45.8
SY 1 SY 5	54.9	34.8	49.2	62.4	47.5	50.5	66.9	53.8	52.5	71.4	66.2	69.2
	10.5	5.4	23.7	25.6	13.4	27.8	32.3	25.4	33.1	49.6	29.1	49.5
VA 1 VA 5 VA10	62.4	40.5	64.9	69.9	48.2	67.6	72.2	52.2	71.2	85.0	69.6	76.9
	54.9	57.5	51.2	63.9	60.9	60.9	66.2	61.5	63.9	69.9	65.2	77.9
	1.5	2.5	22.7	24.8	17.1	31.4	33.1	19.7	32.8	62.4	20.4	42.5

BA—benzoic acid, SA—salicylic acid, SY—syringic acid, VA—vanillic acid, L—lignin, C—cellulose, and H—hemicellulose.

## Data Availability

Not applicable.

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
