# Peer review of "Deciphering the Physicochemical and Microscopical Changes in Ganoderma boninense-Infected Oil Palm Woodblocks under the Influence of Phenolic Compounds"

_plants, 2021, doi:10.3390/plants10091797_

Round 1

Reviewer 1 Report

The manuscript plants-1305233 investigated the potential of plant phenolic compounds (namely benzoic, salicylic, syringic and vanillic acids) to act as antifungal agents against Ganoderma boninense that colonizes root tissues in oil palm (Elaeis guineensis Jacq.).  In vitro studies were undertaken to evaluate the effects of phenolic compounds on the growth of G. boninense in oil palm wood blocks by means of microscopic, spectroscopic and thermodynamics (SEM, TGA and FTIR). The study provides useful information on the influence of investigated phenolics towards both surface morphology and chemistry of wood samples during their degradation by G. boninense.

The manuscript needs to be checked by a native English speaker, as it contains spelling and grammar errors that truly make difficult the reading throughout the entire manuscript.

I have several comments on the manuscript, as follows:

Introduction:

Please rephrase the following:

Page 2, Lines 72-73: ” These naturally occurring phenolic compounds are involved in the lignin synthesis pathway and could be used to eliminate the, G. boninense”.

Material and methods:

Treatment - antifungal activity assays:

  1. Please provide information on the preparation of stock solutions of samples for in vitro studies (solvent type, highest concentration of solvent in culture media).
  2. Authors should consider using a reference drug in their assays.

Conclusions:

Please rephrase: “As a warning, it is essential to determine the threshold concentration of phenolic compounds because the lower concentration has enhanced the colonization of G. boninense which reverses the aim”.

Author Response

We appreciate the comments from the reviewer.  Thank you for your positive comments. We have amended the suggestions given by you. The revised manuscript has been thoroughly checked for grammatical errors. Please see the attachment for point by point response.

Reviewer 2 Report

Dear Authors,

Your manuscript has interesting results. I think that the results you have obtained would have an application primarily in agriculture.
The introduction is well written, easy to read and supported by relevant references, but there are a few self-citations. Could the authors replace the self-citations?
The methods are well described and can be replicated by researchers.
On line 144 I could not understand what this means 11 I asked the same question for line 273. Please explain the authors. The authors can find my question in the attachment. There are some typos that I also noted.
In my opinion, for the greater clarity of the reader, the authors should add "Discussion" to each section of the "Results" section. For example 3.1. Mass loss authors to add a discussion to make it clearer.
I did not see anywhere in the manuscript a separate part "Conclusions". The conclusions are included in the discussion, but in my opinion they should be separated in part conclusions.
Please let the authors carefully review the "instructions for authors" of Journal Plants again and follow them.

Author Response

We appreciate the comments from the reviewer.  Thank you for your positive comments. We have amended the suggestions given by you. The revised manuscript has been thoroughly checked for grammatical errors. Please see the attachment for point by point response and corrections made.

Reviewer 3 Report

The manuscript entitled “Deciphering the physicochemical and microscopical changes in Ganoderma boninense infected oil palm wood blocks under the influence of phenolic compounds” by Surendran et al. intends to screen the efficacy of 4 phenolic compounds on the treatment of Basal Stem Rot disease caused by Ganoderma boninense. The manuscript is very interesting and in the scopus of the journal. the introduction provides a sufficient background, and the material and methods section are well detailed. Also, the discussion is well detailed. In my opinion the drawback of this manuscript is the English, that should be improved. also, future perspectives of studies should be more discussed in the last paragraph.

minor concerns:

- line 74: remove first comma

- remove extra spaces along the manuscript.

Author Response

We appreciate the comments from the reviewer.  Thank you for your positive comments. We have amended the suggestions given by you. and are as follows

SNo.

Comments

Response

1

line 74: remove first comma

The comma has been removed (line 89).

2

remove extra spaces along the manuscript.

The extra spaces have been removed throughout the manuscript.

Thank you very much.

Round 2

Reviewer 2 Report

Dear Authors,

I have read your revised manuscript carefully. I agree to your manuscript being published.